# VTC-LFC: Vision Transformer Compression with Low-Frequency Components

**Zhenyu Wang**
Alibaba Group
`daner.wzy@alibaba-inc.com`

**Hao Luo** *
Alibaba Group
`michuan.lh@alibaba-inc.com`

**Pichao Wang**
Alibaba Group
`pichao.wang@alibaba-inc.com`

**Feng Ding**
Alibaba Group
`dingfeng.dingfeng@alibaba-inc.com`

**Fan Wang**
Alibaba Group
`fan.w@alibaba-inc.com`

**Hao Li**
Alibaba Group
`lihao.lh@alibaba-inc.com`

## Abstract

Although Vision transformers (ViTs) have recently dominated many vision tasks, deploying ViT models on resource-limited devices remains a challenging problem. To address such a challenge, several methods have been proposed to compress ViTs. Most of them borrow experience in convolutional neural networks (CNNs) and mainly focus on the spatial domain. However, the compression only in the spatial domain suffers from a dramatic performance drop without fine-tuning and is not robust to noise, as the noise in the spatial domain can easily confuse the pruning criteria, leading to some parameters/channels being pruned incorrectly. Inspired by recent findings that self-attention is a low-pass filter and low-frequency signals/components are more informative to ViTs, this paper proposes compressing ViTs with low-frequency components. Two metrics named low-frequency sensitivity (LFS) and low-frequency energy (LFE) are proposed for better channel pruning and token pruning. Additionally, a bottom-up cascade pruning scheme is applied to compress different dimensions jointly. Extensive experiments demonstrate that the proposed method could save $40\% \sim 60\%$ of the FLOPs in ViTs, thus significantly increasing the throughput on practical devices with less than $1\%$ performance drop on ImageNet-1K. Code will be available at `https://github.com/Daner-Wang/VTC-LFC.git`.

## 1 Introduction

Recently, Vision transformer (ViT) [14] and its variants [50, 32, 62] have outperformed convolutional neural networks (CNNs) in several vision tasks. However, ViT models still face the challenge of high computational cost when deployed to resource-limited devices. Following previous experiences in compressing CNN models, some pruning methods based on sparse learning [68, 61], taylor expansion [60], or automatic searching [9] have been proposed for ViT models to reduce model redundancy via channel pruning. In addition to the redundancy in parameters, recent literature [48, 28, 41] further points out that some noise tokens mainly encoded task-irrelevant information (*e.g.*, background), and some tokens become similar in deeper layers, showing that great redundancy also exists in tokens.

---

*Corresponding author. This work was done when Zhenyu Wang was an intern at Alibaba.

36th Conference on Neural Information Processing Systems (NeurIPS 2022).

The mainstream works [28, 47, 36] filter out the less informative tokens to reduce the FLOPs without changing the model structure.

Although aforementioned methods have made great progress in ViT compression in spatial domain, we find that they generally suffer from the following two problems: (i) different from CNN pruning which maintains the performance well without finetuning [34, 13], dramatic performance drop is observed when the same method is applied in ViT pruning; (ii) conducting ViT pruning only in spatial domain is not robust to noise, and as shown in Figure 1, after adding noise in the images, the accuracy of spatial compression dramatically drops. To make ViT compression more effective and robust, we propose to conduct ViT compress with the help of frequency domain. Recent studies [3, 55, 42, 38, 52] have indicated that self-attention (SA) behaves like a low-pass filter, and low-frequency signals/components are more informative to ViT models. Inspired by such low-frequency characteristics of ViT, we propose a compression framework named Vision Transformer Compression with Low-Frequency Components (VTC-LFC) which solves the problem from a new angle and emphasizes the contributions of low-frequency components during compression. To our best knowledge, this is the first work that compresses vision transformers in the frequency domain. The main contributions of this paper are listed as below:

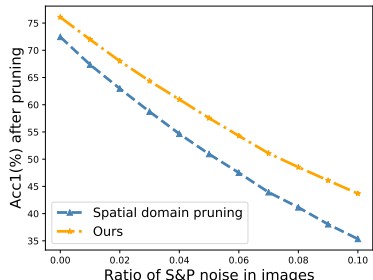

Figure 1: Noise resistance of spatial domain pruning and our pruning. 'S&P' means salt-and-pepper noise, the pruned model is DeiT-Small, and the performance is evaluated on ImageNet-1k.

**Channel pruning based on low-frequency sensitivity:** Channel pruning is a popular structured pruning strategy that aims to remove redundant parameters in fully connected layers of ViT. The mainstream works use some evaluation metrics (*e.g.* Taylor scores [60], weights norm [61], or sparse factor [68]) to estimate importance scores of parameters. Recent studies [3, 42, 52] find that ViTs are more reliable to the low-frequency components in images, *i.e.* the low-frequency information is more important for ViTs. Therefore, we infer that channels that are less effective in encoding low-frequency components will contribute less to the feature representation for ViT models. Motivated by such a property, we propose a better channel pruning criterion named low-frequency sensitivity (LFS) based on the Taylor scores [60]. Different from the standard Taylor scores which are computed with the original images, LFS filters out high-frequency components from images and uses only low-frequency components to estimate the importance of model parameters. In this way, channels that efficiently encode low-frequency information are more likely to be preserved, and the compressed model tends to be more robust to noise. Experimental results show that LFS can alleviate the performance drop after compression without bells and whistles. **Token pruning based on low-frequency energy:** Token compression/sampling aims to select the informative tokens that store more useful information. The popular methods dynamically select those tokens with high correlation to other tokens (*e.g.* the CLS token) as the informative tokens. However, it may be sub-optimal because the selected tokens tend to be similar to each other, and the information included in the token itself has been neglected to some extent. As pointed out by [55, 38], the self-attention module in ViTs behaves like a low-pass filter, *i.e.* the tokens with more low-frequency components can pass more information to the next layers. Inspired by this, we propose improving the attention-based token selection with an extra item, token low-frequency energy (LFE), which quantifies the low-frequency information in tokens. By correcting the attention scores with LFE of tokens, the selector can better distinguish informative tokens from both the long-term dependency in spatial domain and low-frequency contributions in frequency domain.

**Bottom-up Cascade Pruning Framework:** To jointly compress channels and tokens of vision transformers, we propose a bottom-up cascade pruning framework. The model accuracy is further preserved through automatically balancing compression ratios of channel pruning and token pruning block-by-block.

## 2   Related work

**Vision Transformer.** Inspired by the success of transformers [51] in NLP, the Vision Transformer (ViT) [14] is proposed to encode an image into a sequence of tokens and feed them into the pure

transformer architecture. Several studies have shown that ViT performs better than convolution neural networks (CNN) on image classification benchmarks [14, 20] when sufficient training data is provided. Many follow-up variants of ViT [4, 7, 1, 19, 54, 10, 62, 53, 66] have also been proposed. For example, DeiT [50] introduces a distillation token structure into ViT, and LV-ViT [24] proposes the token labeling approach for better training of ViT. In addition to image classification, ViT has also achieved great performance in many other computer vision applications, such as semantic segmentation [11, 56, 12], image retrieval [22, 17], object detection [2, 69] and image reconstruction [8, 59]. However, despite of the outstanding performance in a series of tasks, its high computational cost restricts the deployment of ViT, which motivates the study of lightweight ViT models, including pruning [9, 68, 44, 58], block-weights sharing [26, 64], fast distillation [57], and dynamic prediction architecture [67, 45], among which pruning is a universal approach for almost all model structures.

**ViT Pruning.** As an efficient compression approach, pruning [35, 23, 29, 30, 65, 33, 31] has been widely applied on various convolutional neural networks (CNNs) in computer vision. Pruning approaches [63, 21, 58, 48] have also been proposed for ViT to reduce its model size and inference time. These methods can be roughly grouped into two categories: **1) Channel pruning**, which reduces the number of weights, channels, heads or blocks in ViT. SViTE [9] jointly optimizes parameters and explores connectivity for both unstructured pruning (zeroing weights) and structured pruning (removing heads and channels). ViT-Slim [6] applies $L_1$ sparsity on channels and produces compressed ViTs with unstructured heads (the shape of heads is different). UVC [61] drops heads, channels, and blocks in a unified framework to achieve a high compression ratio. VTP [68] transfers the sparse-learning scheme in CNN pruning to compress ViT. NViT [60] generates smaller networks from the DeiT-base with the Taylor-based pruning scheme. **2) Token pruning**, which focuses on dynamically selecting significant tokens for different inputs. Token pruning would significantly reduce the computational cost while maintaining all parameters. TokenLearner [44] adaptively generates a small set of token vectors according to the spatial attention. EViT [28] downsamples tokens every three blocks and selects tokens with high correlation with the CLS token. DynamicViT [41] estimates the importance of tokens with an MLP [51] based predictor. IA-RED$^2$ [36] introduces a multi-head interpreter to drop uninformative tokens. SP-ViT [25] softly prunes tokens with token selector modules and packages the redundant tokens into one. Different from previous methods, this paper compresses ViTs from a novel prospect, frequency domain, to prune both parameters and tokens in a unified framework.

**Frequency domain analysis for ViT and CNN.** The recent explorations [3, 55, 42, 38, 52] of ViTs have indicated that ViTs behave in an opposite way to CNNs in frequency domain. [3] finds out that ViTs perform better than CNNs when only low-frequency components of images are fed into the models, and proposes the HAT method to enhance the capability of ViTs in capturing high-frequency information. [55] analyzes ViT features from the Fourier spectrum domain and shows that the self-attention module amounts to a low-pass filter. [38] also demonstrates that multi-head self-attentions exhibit opposite behaviors to convolutions, and take advantage of both mechanisms to design a novel AlterNet. To summarize, all these studies point out that the low-frequency components play an important role in information extraction of ViT.

## 3 Methodology

### 3.1 Preliminary

The necessary notations are defined as below. As shown in Figure 2, a transformer block contains a multi-head self-attention (MHSA) module with multiple heads and a feed-forward network (FFN) module with two fully-connection layers. The input images in a mini-batch are denoted as $X \in R^{B \times 3 \times H \times W}$, where $B$, $W$ and $H$ are the batch size, width and height of images, respectively. The inputs of MHSA and FFN in the $l$-th block are denoted as $X^{l,1} \in R^{B \times N^l \times D}$ and $X^{l,2} \in R^{B \times N^l \times D}$, respectively. $N^l$ is the number of tokens, and $D$ is the dimension of a token. In the $l$-th block, the linear projection matrices $W_q^{l,h}$, $W_k^{l,h}$, and $W_v^{l,h}$ are used to calculate $Q_{l,h}$ (query), $K_{l,h}$ (key), and $V_{l,h}$ (value) for the $h$-th attention head. The parameters of the linear projection module in MHSA are denoted as $W_{proj}^l$, and two linear projection matrices in FFN are $W_{fc1}^l$ and $W_{fc2}^l$. Our goal is to reduce the channel number of linear projection matrices and the token number $N^l$.

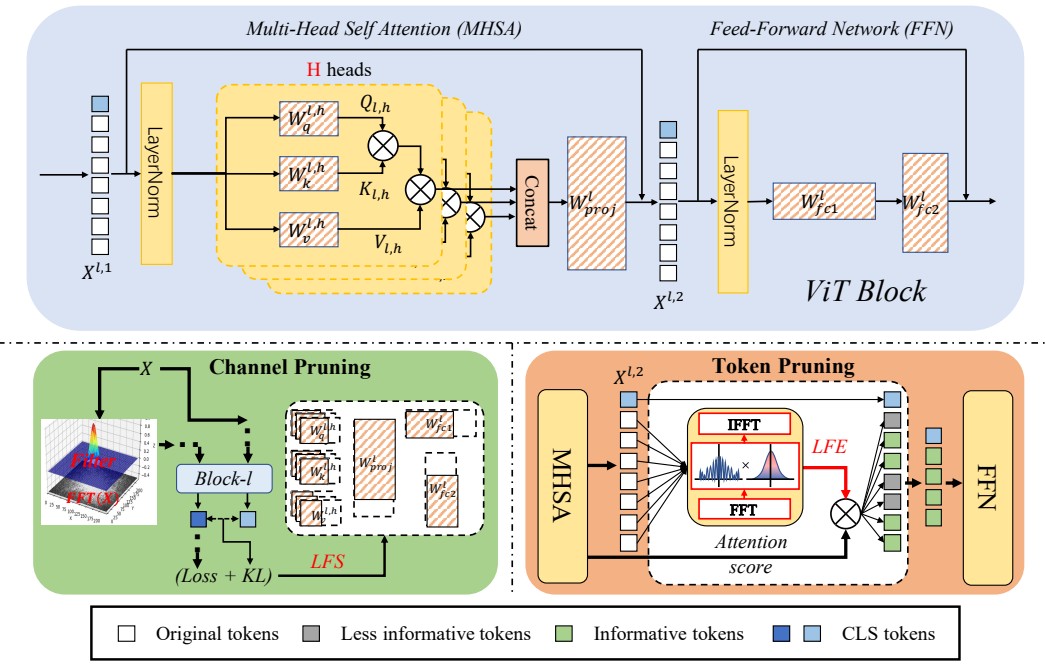

Figure 2: Pruning of channels and tokens in one block. '*LFE*' is the low-frequency energy extracted from tokens according to Equation 6. '*LFS*' denotes the low-frequency sensitivity used to evaluate the importance of channels.

## 3.2 Channel Pruning based on Low-Frequency Sensitivity

As previously introduced, low-frequency components in images are more valuable for the feature representation in ViT models, *i.e.* channels less effective in encoding low-frequency components will contribute less to the feature representation. Therefore, the key goal is to estimate the sensitivity of a channel to low-frequency components in images. To achieve this, we propose an evaluation policy named low-frequency sensitivity (LFS) that estimates the importance scores of model parameters by taking more low-frequency components in images into account.

Assume redundant channels have less influence on model outputs, removing redundant channels should hardly change the value of loss when feeding a set of training images into the model for loss computation. Thus, the importance of a channel can be quantified by the difference in loss induced by removing this channel. Given a number of images $X \in R^{B \times 3 \times H \times W}$ randomly sampled from the training dataset $\mathcal{D}$, the importance score $\mathcal{I}_j$ of a weight $w_j$ is formulated as:

$$\mathcal{I}_j = (\mathcal{L}(\mathcal{M}(X, \mathbf{W}), Y \mid w_j = 0) - \mathcal{L}(\mathcal{M}(X, \mathbf{W}), Y))^2, \tag{1}$$

where $Y \in R^{B \times 1}$ is the label set of data $X$, $\mathcal{L}(\cdot)$ denotes the loss function (cross-entropy loss in this paper), $\mathcal{M}(X, \mathbf{W})$ is the model output, and $\mathbf{W}$ indicates all model weights.

However, the score in Equation 1 can only reflect the importance on the original whole images. To separate low-frequency components from images, low-pass filtering is applied on images in the Fourier spectrum domain before feeding them into ViTs. The low-frequency components in images $\tilde{X}$ are formulated as:

$$\tilde{X} = \mathcal{F}^{-1}(\mathcal{G}(\sigma_c) \odot \mathcal{F}(X)), \tag{2}$$

where $\mathcal{F}(\cdot)$ and $\mathcal{F}^{-1}(\cdot)$ denote the fast Fourier transformation (FFT) [40] and the inverse fast Fourier transform (IFFT), respectively, $\odot$ is the Hadamard product, $\mathcal{G}(\cdot)$ is the low-pass filter, and $\sigma_c \in (0, 1)$ determines the cutoff frequency of the low-pass filter which is similar to the radial averaging of the 2D Fourier spectrum as in [16, 15, 5, 46]. Considering that a binary filter will cause the Ringing effect when the image is transformed back to the spatial domain, Gaussian filter is chosen for $\mathcal{G}(\cdot)$.

In addition to the task-specific loss, the pruned model shall also provide robust feature representation as the original model. In other words, the feature representation of the low-frequency images $\tilde{X}$ shall be as close to that of the original images $X$ as possible. Hence, apart from the cross-entropy loss $\mathcal{L}$ for the classification task, a knowledge-distillation loss is also taken into account. Kullback–Leibler (KL) divergence loss $\mathcal{KL}(\cdot)$ is used to measure the error between the CLS tokens corresponding to the low-frequency image and nature image, respectively. Denote the two CLS tokens as $\tilde{T}$ (from low-frequency images) and $T$ (from nature images), and simplify the cross-entropy loss $\mathcal{L}(\mathcal{M}(X, \mathbf{W}), Y)$ to $\mathcal{L}(X)$. Then, the final importance score $s_j$ of weight $w_j$, named **Low-Frequency Sensitivity** (LFS), is formulated as:

$$s_j = \lambda \cdot \left( \mathcal{L}(\tilde{X} \mid w_j = 0) - \mathcal{L}(\tilde{X}) \right)^2 + (1 - \lambda) \cdot \left( \mathcal{KL}(\tilde{T}, T \mid w_j = 0) - \mathcal{KL}(\tilde{T}, T) \right)^2, \quad (3)$$

where $\lambda$ is the hyper-parameter for the balance of two loss functions.

Calculating the LFS for each parameter with Equation 3 is infeasible for models with millions of parameters. Fortunately, the score can be approximated with the first-order Taylor expansion [34, 60]. Therefore, the approximated version of LFS is represented as below:

$$\hat{s}_j = \lambda \cdot \left( \frac{\partial \mathcal{L}(\tilde{X})}{\partial w_j} \cdot w_j \right)^2 + (1 - \lambda) \cdot \left( \frac{\partial \mathcal{KL}(\tilde{T}, T)}{\partial w_j} \cdot w_j \right)^2, \quad (4)$$

where the gradient terms can be easily obtained in the backward procedure of the model. The channel importance score can then be approximated by summing over LFS scores of all parameters in the channel, *i.e.* the LFS of a channel is computed by the sum of $\hat{s}_j$:

$$\hat{\mathcal{S}}_{\mathcal{J}} = \sum_{j \in \mathcal{J}} \hat{s}_j, \quad (5)$$

where $\mathcal{J}$ means the index set of weights in a channel.

### 3.3 Token Pruning based on Low-Frequency Energy

Token redundancy is another major issue in the ViT compression, and several methods [47, 28, 25] sample informative tokens via analyzing the relationship or attention scores between tokens. Such a solution is sub-optimal because the selected tokens tend to be similar, and the information included in the token itself has been neglected to some extent. To address this problem, the **Low-Frequency Energy** (LFE) is proposed to make use of the low-frequency preference of ViT for token pruning. Following other works [47, 28, 25], as shown in Figure 2, the selector is located between the multi-head self-attention module and the feed-forward network module. Inspired by [55], we evaluate the low-frequency ratio of the token after transforming tokens $X^{l,2}$ into the frequency domain by applying FFT on each channel of tokens, denoted as $\mathcal{X}_{b,:,j}^{l,2} = \mathcal{F}\left( X_{b,:,j}^{l,2} \right)$. We then quantify the low-frequency information contained in a token by calculating the ratio of its remaining energy to the total energy after low-pass filtering. Given filter $\mathcal{G}$ with cutoff factor $\sigma_t$, the LFE is formulated as:

$$\eta_{l,i} = \frac{\left\| \mathcal{LC}\left[ \mathcal{X}^{l,2} \right] \right\|_2}{\left\| \mathcal{DC}\left[ \mathcal{X}^{l,2} \right] \right\|_2} = \frac{\left\| \mathcal{F}^{-1}(\mathcal{G}(\sigma_t) \odot \mathcal{X}_{b,i,:}^{l,2}) \right\|_2}{\left\| \mathcal{F}^{-1}\left( \mathcal{X}_{b,:,:}^{l,2} \right) \right\|_2} = \frac{\left\| \tilde{X}_{b,i,:}^{l,2} \right\|_2}{\left\| X_{b,:,:}^{l,2} \right\|_2}, \quad (6)$$

where $\mathcal{DC}[\cdot]$ and $\mathcal{LC}[\cdot]$ denote the direct-current component and the low-frequency component, respectively. Intuitively, a token with more low-frequency components will achieve a larger $\eta_{l,i}$.

Similar to EViT [28], the attention scores in the spatial domain is also included to evaluate the final importance scores of tokens. For the $h$-th head in ViT, the attention value is calculated as:

$$\mathcal{A}^{l,h} = softmax\left( \frac{Q_{l,h} K_{l,h}^T}{\sqrt{d_{l,h}}} \right), \quad (7)$$

where $\mathcal{A}^{l,h}$ is the attention score matrix and $d_{l,h}$ is the output dimension. The CLS token plays a more significant role than other tokens because it is the final output feature which collects information

from all tokens. Moreover, the head with denser and larger attention values is more important, *i.e.*, with a larger proportion. Thus, our proposed modified attention score is formulated as:

$$\hat{\mathcal{T}}_{l,i} = \frac{1}{H} \sum_{h=0}^{H-1} \left( \theta_{h,0} \cdot \mathcal{A}_{i,0}^{l,h} + \theta_{h,1} \cdot \frac{1}{N^l} \sum_{j=1}^{N^l-1} \mathcal{A}_{i,j}^{l,h} \right), \tag{8}$$

where $\theta_{h,0} = \sum_{j=1}^{N-1} \mathcal{A}_{0,j}^{l,h}$ and $\theta_{h,1} = \mathcal{A}_{0,0}^{l,h}$ are the head-weights of the CLS attention value $\mathcal{A}_{i,0}^{l,h}$ and the other attention value $\mathcal{A}_{i,j}^{l,h}$, respectively.

To estimate the importance score of tokens from multiple and diverse aspects, we consider to combine the LFE $\eta_{l,i}$ and attention score $\hat{\mathcal{T}}_{l,i}$ to get the final importance score of a token as:

$$\tilde{\mathcal{T}}_{l,i} = \hat{\mathcal{T}}_{l,i} \cdot \eta_{l,i}, \tag{9}$$

**Note:** the CLS token is the final output of the ViT model, and is not involved in the token pruning.

### 3.4 Bottom-up Cascade Pruning

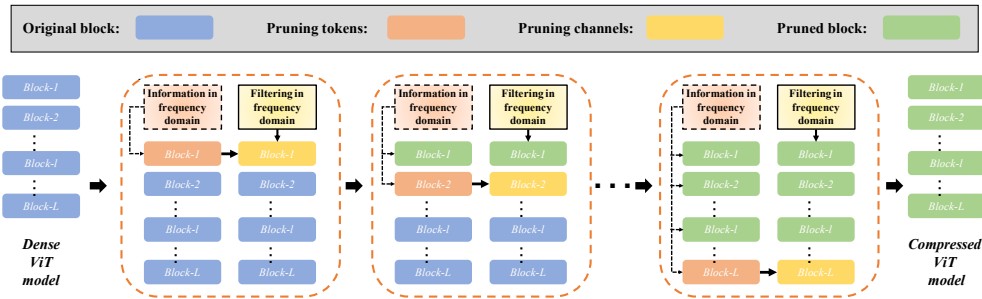

Figure 3: Pipeline of Bottom-up Cascade Pruning. The pruning starts from the first block to the last one. In each block, the token is pruned first according to the information in the frequency domain (LFE criterion). Once token pruning is done, channels will be compressed based on the LFS criterion.

In our method, the value of LFS and LFE in each block is related to the outputs from previous blocks. Hence, the compression in one block will influence the LFS and LFE in its subsequent blocks. It is sub-optimal to independently determine the pruning ratios and indices of channels and tokens for all blocks at once and ignore their inter-relationship. Therefore, we design a **Bottom-up Cascade Pruning** (BCP) process (Figure 3), which promotes pruning from the first block to the last block. A hyper-parameter, named global allowable drop $\varepsilon$, is set to control the final performance drop after pruning. During compression of each block, the number of tokens is gradually reduced until the performance drop reaches $\varepsilon_t = \rho \cdot \varepsilon / L$, where $\rho$ is a hyper-parameter to control the accuracy drop caused by token pruning. Once token pruning is done, channels will be compressed with a similar procedure. When the performance drop caused by pruning reaches $\varepsilon / L$, the compression for a block is considered as completed. More details are described in Algorithm 1 in Appendix A.1.

## 4 Experiments

In this section, the proposed method is evaluated on the benchmark ImageNet (ILSVRC2012) [43], which is a large dataset containing 1.2M training images and 50k validation images of 1000 classes. All the experiments are deployed with Pytorch [39] on NVIDIA V100 GPUs. The code is modified based on the previous study DeiT[2]. The float operations (FLOPs) of models are evaluated by fvcore[3].

---

[2]https://github.com/facebookresearch/deit
[3]https://github.com/facebookresearch/fvcore

## 4.1 Experiments on ImageNet

**Implementation details.** The proposed method is applied to popular ViT models of three different sizes, DeiT-Tiny, DeiT-Small, and DeiT-Base. The latest state-of-the-art (SOTA) methods are compared, including SCOP [49], CP-ViT [47], PoWER [18], HVT [37], IA-RED$^2$ [36], S$^2$ViTE [9], EViT [28], and SPViT [21]. In the pruning procedure, the number of training samples used for evaluating the performance drop in BCP is 5000 (randomly sampling 5 training samples from each category), the number of training samples for calculating LFS is 2000, and the cutoff factors $\sigma_c$ and $\sigma_t$ are 0.1 and 0.85. For three models, DeiT-Tiny, DeiT-Small, and DeiT-Base, the global allowable drop $\varepsilon$ are 9.5, 14, and 14, and the ratio $\rho$ for the allowable drop is 0.56, 0.35, and 0.3 respectively. The removed channels involve the columns (output channels) of $W_q^{l,h}$, $W_q^{l,h}$, $W_q^{l,h}$, and $W_{fc1}^l$ and rows (input channels) of $W_{proj}^l$, and $W_{fc2}^l$. After pruning, the compressed models are fine-tuned with hard distillation [50] of their corresponding original models. The base learning rate is set to 0.0001, and most of the other hyper-parameters follow the settings in [9]. We fine-tune the pruned DeiT-Tiny/DeiT-Small/DeiT-Base models for 300/150/150 epochs. More detailed settings and results of different epochs are listed in Appendix A.3.

**Results and analysis.** The comparison with state-of-the-art methods is shown in Table 1, in which the top-1 accuracy and the reduction ratios of FLOPs are reported. In all three models, the proposed method achieves the highest reduction ratio in FLOPs than previous methods with less than 1% performance drop. Compared to the pure token pruning methods like SCOP [49], CP-ViT [47], PoWER [18], HVT [37], and IA-RED$^2$ [36], our method achieves not only better performance and more reduction in FLOPs, but also less parameters, which demonstrates the superiority of pruning with frequency domain. Although EViT [28], SPViT [21], and S$^2$ViTE [9] achieves better accuracy on DeiT-Base, their reduction in FLOPs (EViT 34.1%, SPViT 33.1%, S$^2$ViTE 33.1% vs Our 57.6%) is much lower than ours. To evaluate the acceleration on inference speed of our pruning technique, the throughput is assessed on a single V100 GPU with batch size 256 in Table 2. The DeiT-Tiny/Small/Base model achieves 69.7%/97.0%/107.1% speed up after pruning, which demonstrates the practicability of the proposed compression approach.

Table 1: Comparison with state-of-the-art methods on ImageNet-1k. 'FLOPs ↓' denotes the reduction ratio of FLOPs. We report two versions with different parameter sizes for our method.

| Method | DeiT-Tiny | | | DeiT-Small | | | DeiT-Base | | |
|---|---|---|---|---|---|---|---|---|---|
| | Top1/Top5(%) | FLOPs ↓ | Params | Top1/Top5(%) | FLOPs ↓ | Params | Top1/Top5(%) | FLOPs ↓ | Params |
| Baseline | 72.2/91.1 | − | 5.7M | 79.8/95.0 | − | 22.1M | 81.8/95.6 | − | 86.4M |
| SCOP [49] | 68.9/− | 38.4% | 5.7M | 77.5/− | 43.6% | 22.1M | 79.7/− | 42.0% | 86.4M |
| PoWER [18] | 69.4/− | 38.4% | 5.7M | 78.3/− | 41.3% | 22.1M | 80.1/− | 39.2% | 86.4M |
| CP-ViT [47] | 71.2/− | 43.3% | 5.7M | 79.1/− | 42.2% | 22.1M | 81.1/− | 41.6% | 86.4M |
| EViT [28] | −/− | − | − | 78.5/94.2 | 50.0% | 22.1M | 81.3/95.3 | 34.1% | 86.4M |
| HVT [37] | 69.7/89.4 | 46.2% | 5.7M | 78.0/93.8 | 47.8% | 22.1M | −/− | − | − |
| IA-RED$^2$ [36] | −/− | − | − | 79.1/94.5 | 31.5% | 22.1M | 80.3/95.0 | 33.0% | 86.4M |
| S$^2$ViTE [9] | 70.1/− | 23.7% | 4.2M | 79.2/− | 31.6% | 14.6M | 82.2/− | 33.1% | 56.8M |
| SPViT [21] | 70.7/90.3 | 23.1% | 4.9M | 78.3/94.3 | 28.3% | 16.4M | 81.6/95.5 | 33.1% | 62.3M |
| **VTC-LFC** | **71.6/90.7** | **46.7%** | **5.1**M | **79.4/94.6** | **54.4%** | **17.7**M | **81.3/95.3** | **57.6%** | **63.5**M |
| **VTC-LFC** | **71.0/90.4** | **41.7%** | **4.2**M | **79.6/94.8** | **47.1%** | **15.3**M | **81.6/95.6** | **54.4%** | **56.8**M |

Table 2: Throughput of baselines and compressed models. 'Speed up' means the improvement in throughput. 'base' denotes the baseline model, and 'pruned' is the compressed model.

| Model | Top1 (base/pruned) | Top5 (base/pruned) | Throughput (base/pruned) | Speed up |
|---|---|---|---|---|
| DeiT-Tiny | 72.2%/71.6% | 91.1%/90.7% | 2648.7/4496.2 | 69.7% |
| DeiT-Small | 79.8%/79.4% | 95.0%/94.6% | 987.9/1946.3 | 97.0% |
| DeiT-Base | 81.8%/81.3% | 95.6%/95.3% | 314.7/651.9 | 107.1% |

Table 3: Results of channel pruning and token pruning with different criteria on DeiT-Small. In the column of 'Acc1 (%)', 'FT' means fine-tuning. 'FLOPs ↓' denotes the reduction ratio of FLOPs. It is noted that the original NViT compresses a large-scale model to the target size (*e.g.* ViT-Small) and uses extra CNN teacher models, so we implement NViT* to compress ViT-Small to the pruned ViT-Small under the standard pruning setting for fair comparison here. For a clear comparison, BCP is not applied in any experiments here.

| Channel Pruning | | Token Pruning | | Acc1 (%) | | FLOPs ↓ |
| NViT*[60] (baseline) | LFS (ours) | EViT[28] (baseline) | LFE (ours) | before FT | after FT | |
| --- | --- | --- | --- | --- | --- | --- |
| × | × | × | × | 79.8 | − | 0.0% |
| ✓ | × | × | × | 47.5 | 78.9 | 32.8% |
| × | ✓ | × | × | 61.2 | 79.4 | 32.8% |
| × | × | ✓ | × | 76.8 | 79.6 | 43.3% |
| × | × | × | ✓ | 77.6 | 80.1 | 43.3% |
| ✓ | × | ✓ | × | 40.5 | 78.0 | 55.0% |
| × | ✓ | × | ✓ | 57.9 | 78.7 | 55.0% |

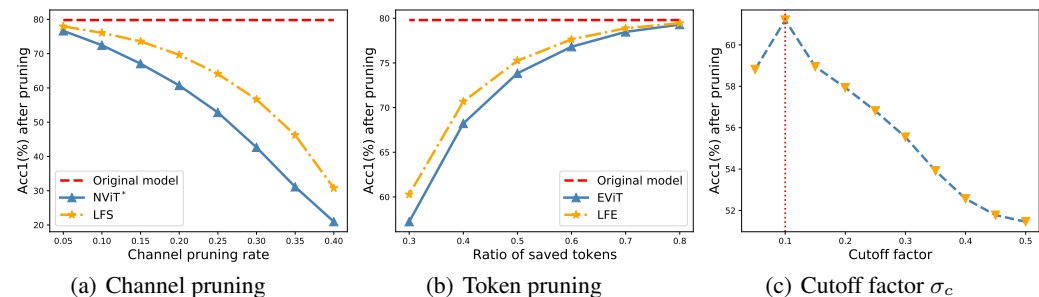

(a) Channel pruning     (b) Token pruning     (c) Cutoff factor $\sigma_c$

Figure 4: (a) results of channel pruning with two different criteria 'NViT*' and 'LFS'. (b) results after token pruning with criteria 'EViT' and 'LFE'. (c) results with different cutoff factors to determine the cutoff frequency of low-pass filters.

## 4.2 Ablation Study

**Effectiveness of LFS and LFE.** For fair comparison to analyze the effectiveness of the proposed LFS (for channel pruning) and LFE (for token pruning), we conduct experiments without using BCP framework in Table 3. Two state-of-the-art methods NViT [60] and EViT [28] are selected as baselines for channel pruning and token pruning, respectively. NViT identifies channels based on the Taylor score and EViT selects tokens according to the attention score. The channels are pruned globally using the manual pruning rate as NViT, and the number of tokens is determined following the same ratio as EViT. The main difference between other methods and ours is whether to leverage the characteristics of ViT on low-frequency components. Table 3 shows the results of only compressing channels, tokens, and both. For channel pruning, it can be found that LFS outperforms NViT* with the same FLOPs reduction. For token pruning, LFE achieves an even higher accuracy than the original DeiT-Small. Both comparison results demonstrate the superiority of pruning based on information in the frequency domain over using only information in the space domain. The results of different pruning ratios for channels and tokens are displayed in Figure 4(a) and 4(b), in which the proposed method consistently outperform the other methods under all ratios. The distributions of importance scores for models are displayed in Appendix A.10 Fig. 5.

The influence of the cutoff factor, which determines the ratio of saved low-frequency components to all frequency, is also analyzed. As shown in Figure 4(c), $\sigma_c = 0.1$ is the sweet spot, which further proves the preference of ViT on low-frequency information. For $\sigma_t$, $0.85$ is the best choice.

Table 4: Automatically searching pruning ratios with BCP.'$\varepsilon$' is the hyper-parameter that controls the performance drop caused by pruning, and '$\rho$' is a hyper-parameter that balances channel pruning and token pruning. All results are achieved by DeiT-Small.

| $\varepsilon/\rho$ | 10/0.35 | 13/0.35 | 14/0.15 | 14/0.35 | 14/0.55 | 15/0.35 | 18/0.35 |
|---|---|---|---|---|---|---|---|
| **FLOPs** $\downarrow$ | 48.6% | 54.0% | 46.0% | 54.4% | 59.2% | 55.2% | 58.5% |
| **Params** | 18.7M | 17.8M | 17.3M | 17.7M | 17.9M | 17.4M | 16.8M |
| **Acc1 w/o FT** | 71.6% | 69.2% | 68.8% | 68.3% | 69.0% | 66.9% | 64.6% |
| **Acc1 w/ FT** | 79.8% | 79.3% | 79.9% | 79.4% | 78.7% | 79.1% | 78.9% |

**Factor analysis for BCP.** The pruning ratios for channels and tokens are determined by two hyper-parameters $\varepsilon$ and $\rho$. As shown in Table 4, when maintaining $\rho = 0.35$ and increasing $\varepsilon$ from 10 to 18, the model parameter size will be reduced from 18.7M to 16.8M, and the performance ranges from 79.8% to 78.9%. Similarly, when maintaining $\varepsilon = 14$ and increasing $\rho$ from 0.15 to 0.55, both the model size and the FLOPS reduction ratio are increased, while the performance is reduced from 79.9% to 78.7%. Compared with the result (78.7% accuracy and 55.0% Flops reduction ratio) without BCP in the last row of Table 3, we can observe that BCP improves the accuracy or compression ratio of the model, which verifies the effectiveness of the proposed BCP strategy. After comprehensively considering the model parameters size, the FLOPS reduction ratio and model accuracy, we set the $\varepsilon = 14$ and $\rho = 0.35$ in this paper. These two parameters can be adjusted according to specific requirements in real-world applications.

**Influence of each module.** To further study how each component affects the proposed method, LFS, LFE, and BCP are respectively removed from our scheme. The modified versions are then applied on the same model. Since BCP will automatically adjust pruning ratios of each block, it is necessary to introduce additional variables for controlled experiments. For a fair comparison, we keep the same pruning ratios (determined by our VTC-LFC) of each block in all experiments. As shown in Table 5, the proposed LFS/LFE/BCP improves the performance by 1.3%/0.6%/0.9% before fine-tuning and 0.1%/0.5%/0.2% after fine-tuning, respectively. The experimental results show the effectiveness of the proposed modules.

Table 5: Effects of LFS, LFE, and BCP on the proposed compression method. 'Original' is the original model without pruning. 'Ours' is the result including LFS, LFE, and BCP. 'NvEv-P' means pruning globally as strategies in NViT and EViT rather than the proposed block-by-block scheme.

| Model | Method | Acc1 (before FT) | Acc1 (after FT) | FLOPs $\downarrow$ |
|---|---|---|---|---|
| | Original | 79.8% | − | 0.0% |
| | VTC-LFC (Ours) | 68.3% | 79.4% | 54.4% |
| DeiT-Small | w/o LFS (LFS→NViT*) | 67.0% | 79.3% | 54.4% |
| | w/o LFE (LFE→EViT) | 67.7% | 78.9% | 54.4% |
| | w/o BCP (BCP→NvEv-P) | 67.4% | 79.2% | 54.4% |

**Influence of pruning sequence.** The main reason for pruning from the first block to the last one is that the pruning in previous blocks will change the inputs of their subsequent blocks, which may influence the value of our proposed LFS and LFE. If the last block is pruned first, the selected channels and tokens in this block will need to be re-adjusted when former blocks are compressed. Instead, if the former blocks are pruned firstly, the inputs for the subsequent blocks are fixed. Both pipelines are executed and compared on DeiT-Small, and the results shown in Table 6 demonstrate the advantage of the proposed buttom-up (from first to last) pipeline.

Table 6: Pruning with different sequences. 'Bottom-up' means pruning from the first block to the last one while 'Top-down' denotes pruning from the last block to the first one.

| Pipeline | Top1/Top5 w/o FT | Top1/Top5 w/ FT | FLOPs | Params |
|---|---|---|---|---|
| Bottom-up (Ours) | 68.3/89.1 | 79.4/94.6 | 2.1G | 17.7M |
| Top-down | 68.2/89.0 | 78.9/94.5 | 2.1G | 17.6M |

**Influence of hyper-parameter $\lambda$.** The hyper-parameter $\lambda$ used to balance different terms in LFS during compression is analyzed here. As listed in Table 7, the models pruned with different $\lambda$ values are evaluated after pruning. It can be found that the model achieves the best accuracy when $\lambda$ is 0.1. In addition, the proposed method is not very sensitive to the value of $\lambda$ ($\lambda > 0$).

Table 7: Pruning 20% channels with different $\lambda$.

| Hyper-parameter $\lambda$ | 0.0 | 0.1 | 0.2 | 0.3 | 0.4 | 0.5 | 0.6 | 0.7 |
|---|---|---|---|---|---|---|---|---|
| Top1 accuracy after pruning | 65.90 | 69.76 | 69.70 | 69.73 | 69.73 | 69.70 | 69.67 | 69.67 |

**Compression on other ViT models.** In addition to DeiT models, the proposed method is also evaluated on LV-ViT [24] and window-based attention model Swin [32], in which the pruned models are fine-tuned for 150 epochs, respectively. Specifically, due to the token downsampling and the window shifting, token pruning is not adapted to Swin yet so that only channel pruning is adopted on Swin. To simplify, the token labels in the original LV-ViT are not used during fine-tuning. For LV-ViT, our method obtains 0.3% higher (81.8% vs 81.5%) performance and 0.1G lower FLOPs (3.2G vs 3.3G) than the combination of existing channel pruning (NViT) and token pruning (EViT) approaches. On Swin, the proposed VTC-LFC achieves 0.2/3.1% higher accuracy than the previous SPViT [21]/STEP [27] with fewer FLOPs (3.3G vs 3.4G/3.5G) and parameters (17.1M vs 25.8M/23.6M). The results shown in Table 8 demonstrate that proposed method can still achieve better performance as well as lower FLOPs than previous approaches on other ViT architectures.

Table 8: Results for LV-ViT and Swin Transformer backbones on the ImageNet-1k.

| Model | Method | Top1(%) | Top5 (%) | FLOPs (G) | Params (M) |
|---|---|---|---|---|---|
| LV-ViT-S | Original | 83.2 | 96.3 | 6.5 | 25.8 |
| | NViT*+EViT | 81.5 | 95.3 | 3.3 | 20.2 |
| | VTC-LFC (Ours) | 81.8 | 95.6 | 3.2 | 20.2 |
| Swin-Tiny | Original | 81.1 | 95.5 | 4.5 | 28.3 |
| | STEP [27] | 77.2 | 93.6 | 3.5 | 23.6 |
| | SPViT [21] | 80.1 | 95.0 | 3.4 | 25.8 |
| | VTC-LFC (Ours) | 80.3 | 95.0 | 3.3 | 17.1 |

# 5 Conclusion and Discussion

This paper reveals the disadvantages of pruning ViTs only in the spatial domain and takes advantage of the preference of ViTs for low-frequency information to conduct compression. Two metrics, low-frequency sensitivity and low-frequency energy are proposed to leverage knowledge in the frequency domain for better channel pruning and token pruning. The comparison with the spatial domain pruning approaches proves that the proposed method can identify the informative channels and tokens more precisely, thus better maintaining the model accuracy. Additionally, with the proposed bottom-up cascade pruning strategy, both channels and tokens are automatically compressed in a unified framework. Extensive experiments of different models on ImageNet demonstrate that more than half of computational costs are saved from ViTs, with significant improvements in inference efficiency. The comparison with the latest compression methods shows the superiority (better performance and more FLOPs reduction) of the proposed approach. As a preliminary study about pruning ViTs with the frequency domain, more efficient ways and deeper studies are expected to be explored based on it.

# 6 Acknowledgement

This work was supported by Alibaba Group through Alibaba Research Intern Program.

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
