# A   Appendix

## A.1   Detailed procedure of bottom-up cascade pruning

The procedure of the proposed bottom-up cascade pruning (BCP) is illustrated here. As shown in Algorithm 1, the BCP greedily prunes channels and tokens block-by-block until the performance drop reaches the default threshold (allowable drop $\varepsilon$).

---

**Algorithm 1** Bottom-up Cascade Pruning.

---

**Input:**
    The pretrained ViT model $\mathcal{M}$ and the number of blocks $L$;
    The global allowable drop $\varepsilon$ and the ratio of token pruning caused drop $\rho$;
    The training set $\mathcal{D}_{train}$;
 1: sample images from $\mathcal{D}_{train}$ as the testing set $\mathcal{D}_{test}$;
 2: respectively set allowable drops for token pruning and channel pruning: $\varepsilon_t = \rho \cdot \varepsilon / L$, $\varepsilon_c = \varepsilon / L$;
 3: get best accuracy with $\psi_{best} = Acc\left(\mathcal{M}, \mathcal{D}_{test}\right)$;
 4: **for** each $l \in [1, L]$ **do**
 5:     set accuracy $\psi_l = 0$, set pruned number of tokens $p = 1$, and get the number of left tokens $n_t^l$;
 6:     **while** $(\psi_{best} - \psi_l < \varepsilon_t) and (p > 0)$ **do**
 7:         set the number of tokens from block $l$ to $L$ as $n_t^l - p$;
 8:         select tokens according to LFE and get compressed model $\mathcal{M}_{prune}$;
 9:         get accuracy with $\psi_l = Acc\left(\mathcal{M}_{prune}, \mathcal{D}_{test}\right)$;
10:         update $p$ with $p = min\left(\lceil(2 - (\psi_{best} - \psi_l)/\varepsilon_t) \cdot p\rceil, \lceil p + n_t^l/3\rceil\right)$;
11:     **end while**
12:     set the number of tokens from block $l$ to $L$ as $n_t^l - p$;
13:     sample images from $\mathcal{D}_{train}$ as set $\mathcal{D}_l$ and get accuracy with $\psi_l = Acc\left(\mathcal{M}_{prune}, \mathcal{D}_{test}\right)$;
14:     get the number of left channels $n_c^l$, and set pruned number of channels $p = \lceil 0.01 \cdot n_c^l \rceil$;
15:     **while** $(\psi_{best} - \psi_l < \varepsilon_c) and (p > 0)$ **do**
16:         get channel low-frequency sensitivity with LFS and set $\mathcal{D}_l$;
17:         remove $p$ channels with the smallest sensitivity from current block;
18:         get accuracy of pruned model: $\psi_l = Acc\left(\mathcal{M}_{prune}, \mathcal{D}_{test}\right)$;
19:         update $p$ with $p = min\left(\lceil(2 - (\psi_{best} - \psi_l)/\varepsilon_t) \cdot p\rceil, \lceil p + (n_c^l - p)/4\rceil, n_c^l\right)$;
20:     **end while**
21:     remove $p$ channels with the smallest sensitivity from current block;
22:     get accuracy of pruned model: $\psi_l = Acc\left(\mathcal{M}_{prune}, \mathcal{D}_{test}\right)$;
23:     update $\psi_{best} = \psi_l$
24: **end for**
**Output:**
    The compressed ViT model, $\mathcal{M}_{prune}$;

---

## A.2   Distribution of selected tokens

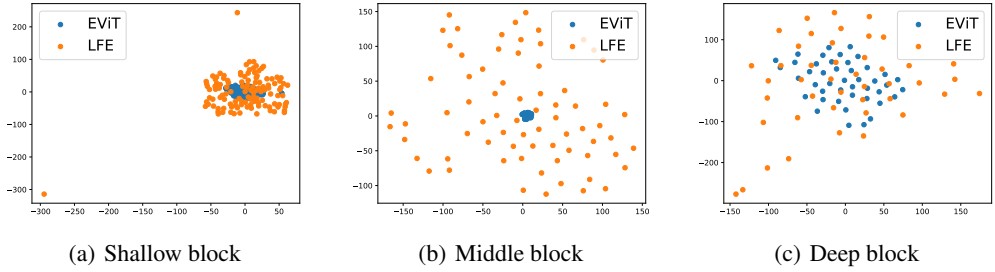

(a) Shallow block        (b) Middle block        (c) Deep block

Figure 1: Distribution of selected tokens. All results are achieved by reducing dimension with TSNE.

In the introduction, we have indicated that selecting tokens only with attention values may maintain many similar tokens. To give a better illustration, the distribution of selected tokens with attention-

based approach (EViT) and our proposed LFE are visualized in Figure 1. it is obviously that our method can obtain diverse tokens rather than similar tokens.

## A.3 Details of pruning and fine-tuning

The detailed hyper-parameters of pruning and fine-tuning are reported in Table 1 and Table 2, so that our work can be reproduced by other researchers. The results of fine-tuning with different epochs are listed in Table 3.

Table 1: Hyper-parameters for pruning.

| model (acc1) | allowable drop $\varepsilon$ | ratio $\rho$ | cutoff factor $\sigma_c, \sigma_t$ |
|---|---|---|---|
| DeiT-Tiny (71.6%) | 9.5 | 0.56 | 0.1, 0.85 |
| DeiT-Small (79.4%) | 14 | 0.35 | 0.1, 0.85 |
| DeiT-Base (81.3%) | 14 | 0.3 | 0.1, 0.85 |

Table 2: Hyper-parameters for fine-tuning.

| config | value |
|---|---|
| optimizer | AdamW |
| base learning rate | 1e-4 |
| weight decay | 0.05 |
| optimizer momentum | $\beta_1, \beta_2$=0.9, 0.999 |
| batch size | 512 (Tiny), 256 (Small), 128 (Base) |
| learning rate schedule | cosine decay |
| warmup epochs | 0 |
| training epochs | 300 (Tiny), 150 (Small), 150 (Base) |
| label smoothing | 0.1 |
| mixup | 0.8 |
| cutmix | 1.0 |
| drop path | 0.1 |
| exp. moving average (EMA) | 0.99996 |
| distillation loss | corss entropy loss |
| distillation-alpha | 0.1 (Tiny), 0.25 (Small/Base) |
| layer-decay | 0.9 |

Table 3: Fine-tuning with different epochs.

| Model | 75 epochs Top1/Top5(%) | 150 epochs Top1/Top5(%) | 300 epochs Top1/Top5(%) | GPU Days (A10) 75ep/150ep/300ep |
|---|---|---|---|---|
| DeiT-Tiny | 70.1/89.9 | 70.8/90.2 | 71.6/90.7 | 2.3/4.6/9.2 |
| DeiT-Small | 78.6/94.4 | 79.4/94.6 | 79.8/94.9 | 3.3/6.7/13.3 |
| DeiT-Base | 81.2/95.2 | 81.3/95.3 | 81.3/95.1 | 7.5/15.0/30.0 |

## A.4 Approximation of LFS

Here, we provide the detailed derivation for the approximated low-frequency sensitivity (LFS) $\hat{s}_j$. Given complete LFS $s_j$ formulated as:

$$s_j = \lambda \cdot \left( \mathcal{L}(\tilde{X} \mid w_j = 0) - \mathcal{L}(\tilde{X}) \right)^2 + (1 - \lambda) \cdot \left( \mathcal{KL}(\tilde{T}, T \mid w_j = 0) - \mathcal{KL}(\tilde{T}, T) \right)^2. \quad (1)$$

By performing the Taylor expansion, the loss functions $\mathcal{L}(\tilde{X} \mid w_j = 0)$ and $\mathcal{KL}(\tilde{T}, T \mid w_j = 0)$ can be rewritten as:

$$\mathcal{L}(\tilde{X} \mid w_j = 0) = \mathcal{L}(\tilde{X}) + \frac{\partial \mathcal{L}(\tilde{X})}{\partial w_j} \cdot w_j + \theta(\mathcal{L}),$$
$$\mathcal{KL}(\tilde{T}, T \mid w_j = 0) = \mathcal{KL}(\tilde{T}, T) + \frac{\partial \mathcal{KL}(\tilde{T}, T)}{\partial w_j} \cdot w_j + \theta(\mathcal{KL}), \quad (2)$$

where $\theta(\mathcal{L})$ and $\theta(\mathcal{KL})$ are the sum of high-order terms. Considering that the first-order expansion has had high correlation to the original loss function and the calculation for the higher-order terms is inefficient, only the zero-order and first-order terms are maintained:

$$
\begin{aligned}
\mathcal{L}(\tilde{X} \mid w_j = 0) &= \mathcal{L}(\tilde{X}) + \frac{\partial \mathcal{L}(\tilde{X})}{\partial w_j} \cdot w_j, \\
\mathcal{KL}(\tilde{T}, T \mid w_j = 0) &= \mathcal{KL}(\tilde{T}, T) + \frac{\partial \mathcal{KL}(\tilde{T}, T)}{\partial w_j} \cdot w_j.
\end{aligned}
\tag{3}
$$

Substituting this expansion into the complete LFS, the approximated LFS $\hat{s}_j$ is formulated as:

$$
\hat{s}_j = \lambda \cdot \left( \frac{\partial \mathcal{L}(\tilde{X})}{\partial w_j} \cdot w_j \right)^2 + (1 - \lambda) \cdot \left( \frac{\partial \mathcal{KL}(\tilde{T}, T)}{\partial w_j} \cdot w_j \right)^2,
\tag{4}
$$

## A.5 Resistance of pruned models to different noises

In addition to the salt-and-pepper noise shown in the introduction, the resistance of models to other noises (including Gaussian noise, exponential noise, and uniform noise) are also evaluated. As shown in Figure 2, for all kinds of noises, the performance of our pruned models drops slower, which demonstrates the robustness of channels selected by our method.

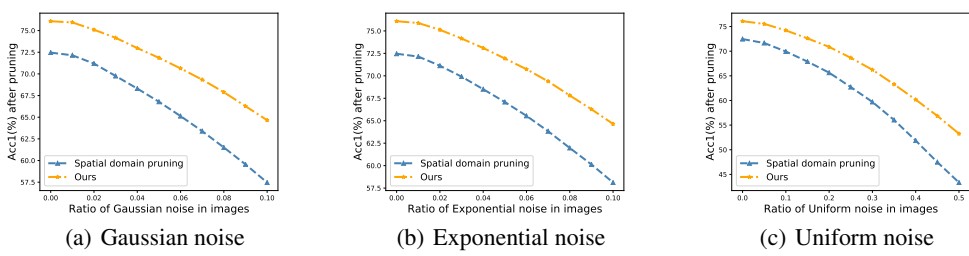

| (a) Gaussian noise | (b) Exponential noise | (c) Uniform noise |

Figure 2: Noise resistance of spatial domain pruning and our pruning.

## A.6 Throughput of different models

Table 4: Throughput of baselines and compressed models. 'Speed up' means the improvement in throughput. 'base' denotes the baseline model, and 'pruned' is the compressed model.

| Model | Acc1 (%) base/pruned | FLOPs (G) base/pruned | Params (M) base/pruned | Throughput base/pruned | Speed up |
|---|---|---|---|---|---|
| DeiT-Tiny | 72.2/71.6 | 1.3/0.67 | 5.7/5.1 | 2648.7/4496.2 | 69.7% |
|  | 72.2/71.0 | 1.3/0.73 | 5.7/4.2 | 2648.7/4021.9 | 51.8% |
| DeiT-Small | 79.8/79.4 | 4.6/2.10 | 22.1/17.7 | 987.9/1946.3 | 97.0% |
|  | 79.8/79.6 | 4.6/2.44 | 22.1/15.3 | 987.9/1617.9 | 63.8% |
| DeiT-Base | 81.8/81.3 | 17.6/7.46 | 86.4/63.5 | 314.7/651.9 | 107.1% |
|  | 81.8/81.6 | 17.6/8.02 | 86.4/56.8 | 314.7/609.3 | 93.6% |

## A.7 Performances on ImageNet-Real and CIFAR-10

Apart from ImageNet-1k, the pruned models are also evaluated on ImageNet-Real and CIFAR-10. For CIFAR-10, the compressed DeiT-Tiny/Small/Base is fine-tuned for 300/150/75 epochs. The results shown in Table 5 demonstrates that the pruned models perform well on other datasets.

## A.8 Spectrum of channels

The spectra of output features corresponding to saved channels and pruned channels are visualized in Figure 3, where the map with green box is saved while the map with red box is pruned. The

Table 5: Performance on ImageNet-Real and CIFAR-10.

| Dataset | Model | Top1 (%) base/pruned | Top5 (%) base/pruned | FLOPs (G) base/pruned | Params (M) base/pruned |
|---|---|---|---|---|---|
| ImageNet-Real | DeiT-Tiny | 80.1/79.5 | 94.4/94.2 | 1.3/0.7 | 5.7/5.1 |
| | DeiT-Small | 85.7/85.4 | 96.9/96.7 | 4.6/2.1 | 22.1/17.7 |
| | DeiT-Base | 86.8/86.2 | 97.1/96.9 | 17.6/7.5 | 86.4/63.5 |
| CIFAR-10 | DeiT-Tiny | 98.1/97.9 | 99.9/99.9 | 1.3/0.7 | 5.5/4.9 |
| | DeiT-Small | 98.6/98.6 | 99.9/99.9 | 4.6/2.1 | 21.7/17.3 |
| | DeiT-Base | 98.9/98.8 | 99.9/99.9 | 17.6/7.5 | 85.8/62.7 |

baseline method (NViT) without taking advantage of characteristic in frequency domain is selected for comparison. Although selection of two methods in channels is similar, our method tends to save the channel with more low-frequency components (row 1, column 4 in Figure 3(b)) while NViT chooses the channel with uniform values (row 2, column 3 in Figure 3(a)). The accumulation of such changes in selection finally result in our better performance, which further proves the importance of low-frequency information to ViTs.

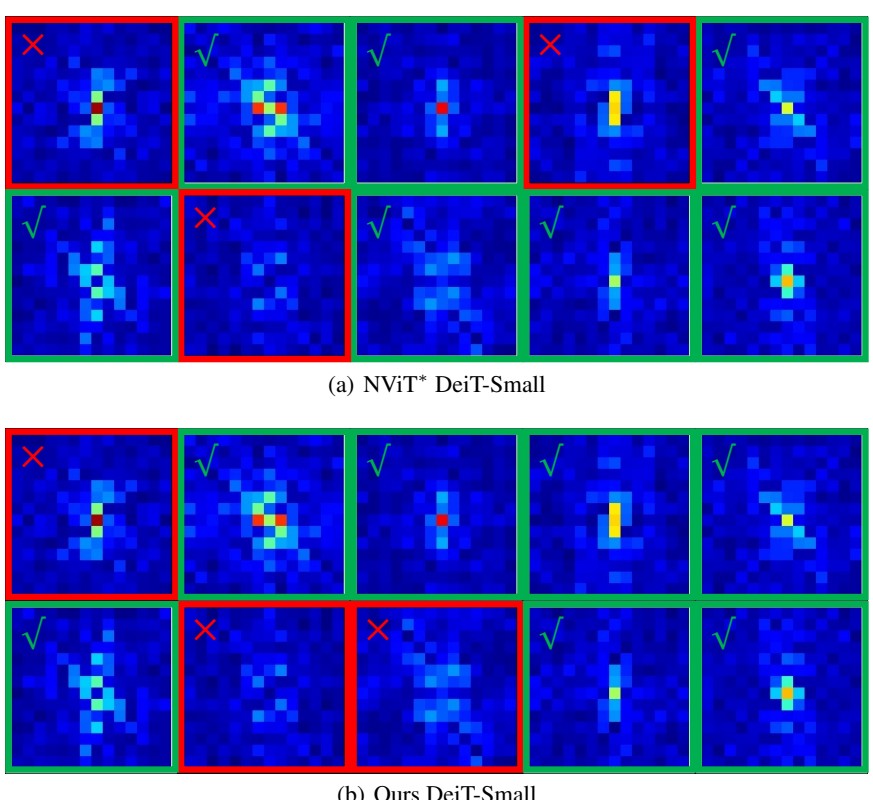

(a) NViT* DeiT-Small

(b) Ours DeiT-Small

Figure 3: Output spectrum of each channel in Fourier domain. The green box means the saved channel, while the red box denotes the pruned channel.

## A.9 Architecture of pruned models

To give a straightforward visualization for the pruned models, the ratios of saved channels and tokens in each block is presented in Figure 4. It can be observed that more channels are pruned than tokens in shallow blocks (*e.g.* block-1 and block-2), while more tokens are reduced in deep blocks. This inspires that the architecture with thin channels in shallow layers and few tokens in deep layers may be more suitable to ViTs.

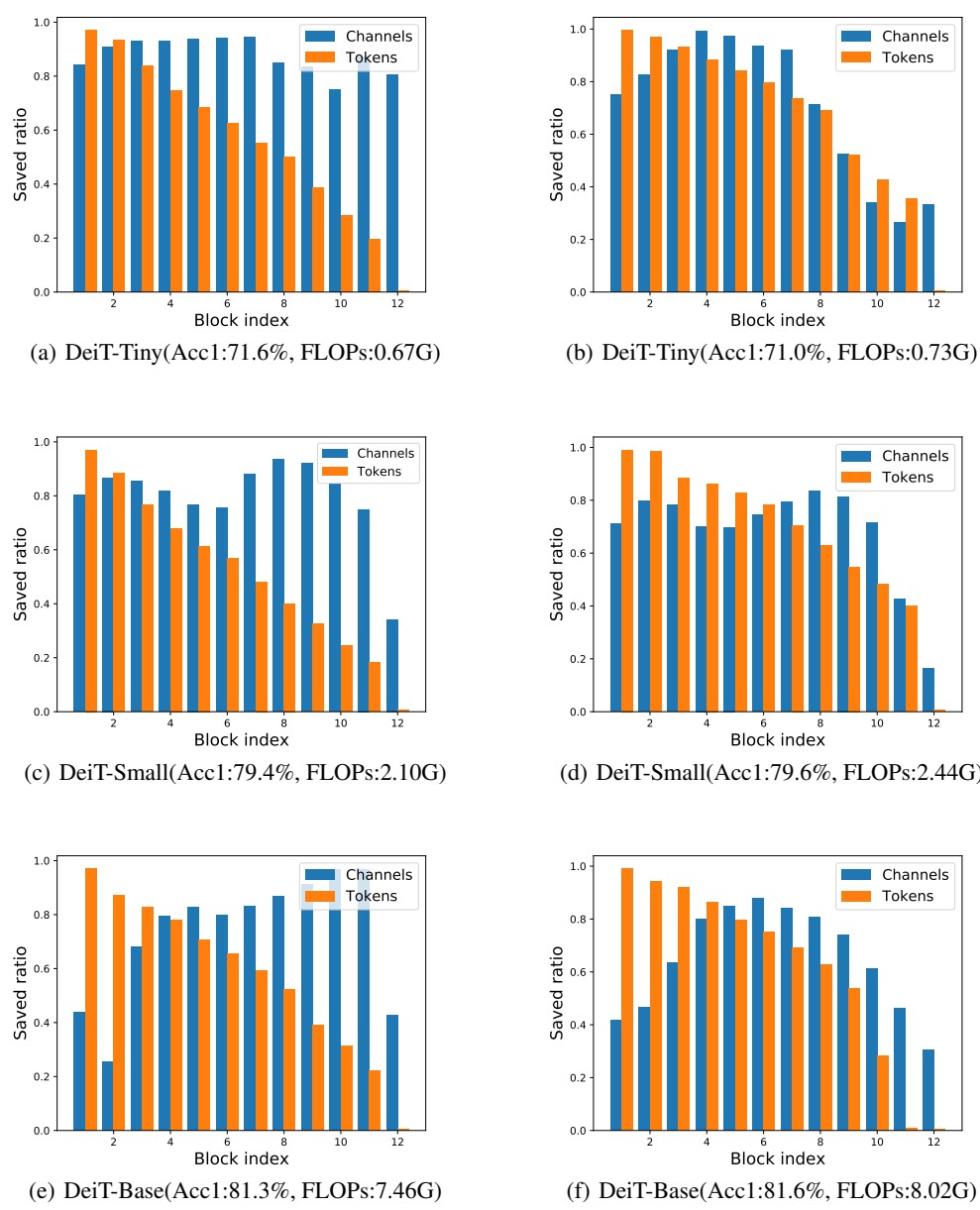

(a) DeiT-Tiny(Acc1:71.6%, FLOPs:0.67G)

(b) DeiT-Tiny(Acc1:71.0%, FLOPs:0.73G)

(c) DeiT-Small(Acc1:79.4%, FLOPs:2.10G)

(d) DeiT-Small(Acc1:79.6%, FLOPs:2.44G)

(e) DeiT-Base(Acc1:81.3%, FLOPs:7.46G)

(f) DeiT-Base(Acc1:81.6%, FLOPs:8.02G)

Figure 4: Architecture of pruned models.

## A.10 Distributions of importance scores

The distributions of the proposed importance scores for channels (LFS) and tokens (LFE×attention-score) are displayed in Fig. 5. 2000 images from ImageNet-1k are sampled and fed into DeiT-Tiny to obtain the average LFS for each channel and scores for tokens. Compared to the original model (Fig. 5(a)), the density of LFS near zero is lower in the pruned model. Similarly, the token scores in the pruned model are denser than those in the original model (Fig. 5(b)).

## A.11 Low-pass filtering on input images

In Figure 6, the images filtered with different cutoff factors are visualized in both the spatial domain and the Fourier domain.

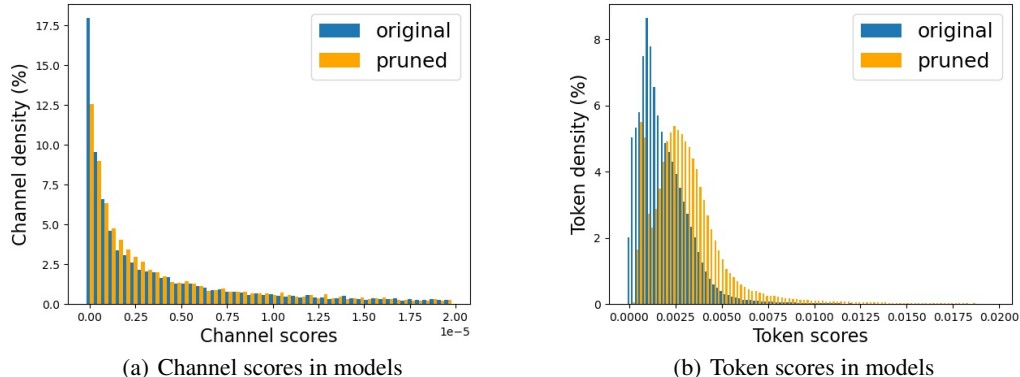

(a) Channel scores in models

(b) Token scores in models

Figure 5: Distributions of importance scores. 'channel scores' denotes LFS (low-frequency sensitivity). 'token scores' means 'LFE (low-frequency energy)×attention-score'.

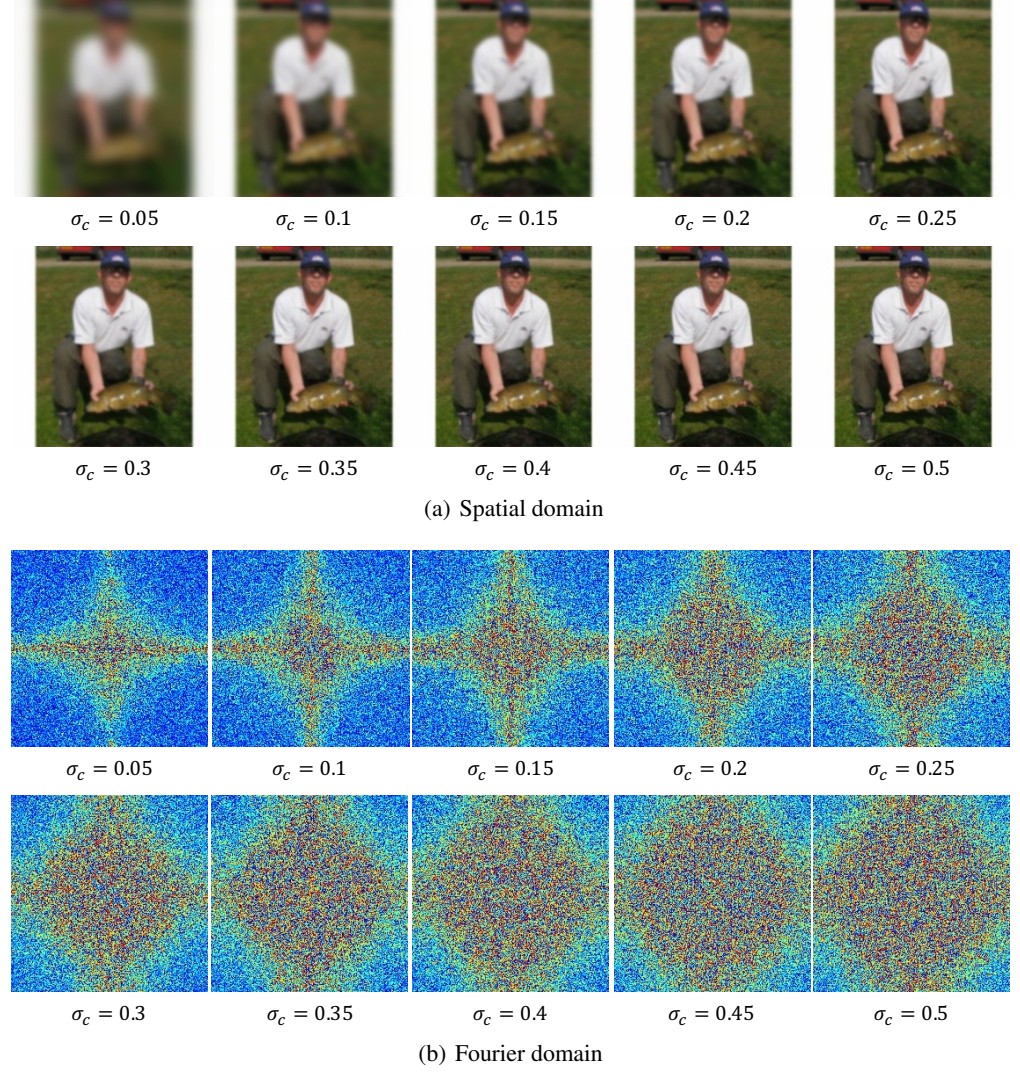

$\sigma_c = 0.05$      $\sigma_c = 0.1$      $\sigma_c = 0.15$      $\sigma_c = 0.2$      $\sigma_c = 0.25$

$\sigma_c = 0.3$      $\sigma_c = 0.35$      $\sigma_c = 0.4$      $\sigma_c = 0.45$      $\sigma_c = 0.5$

(a) Spatial domain

$\sigma_c = 0.05$      $\sigma_c = 0.1$      $\sigma_c = 0.15$      $\sigma_c = 0.2$      $\sigma_c = 0.25$

$\sigma_c = 0.3$      $\sigma_c = 0.35$      $\sigma_c = 0.4$      $\sigma_c = 0.45$      $\sigma_c = 0.5$

(b) Fourier domain

Figure 6: Images after filtering. (a) is the images after low-pass filtering. (b) is the spectrum of images in the Fourier domain.