# OpenReview forum: "VTC-LFC: Vision Transformer Compression with Low-Frequency Components"
_NeurIPS.cc/2022/Conference — NeurIPS 2022 Accept_

### Official Review · Reviewer_2SYW · 2022-07-10

**Rating:** 7
**Confidence:** 3
**Soundness:** 2 fair
**Presentation:** 3 good
**Contribution:** 3 good

**Summary:**

The major contributions of this paper are 2-fold:
1) By leveraging on the understanding regarding the low-frequency information preference of ViTs, this work proposes improved channel pruning and token pruning schemes to compress ViTs. This is achieved using 3 components below:
- **Channel pruning** : A quantitative metric called LFS (Low Frequency Sensitivity) is proposed to estimate the importance of model parameters by largely considering low-frequency components in images. This is used for channel pruning.
- **Token pruning** : A quantitative metric called LFE (Low Frequency Energy) is proposed to estimate the percentage of low-frequency information in tokens.
- **A bottom-up cascaded pruning framework** is proposed to jointly compress channels and tokens of ViTs.

2) The proposed method obtains large improvements in ViT compression (i.e.: reduce FLOPs by 40%-60% at the expense of only 1% top1 accuracy) in ImageNet based classification setups.


**Questions:**

Please see Weaknesses section above for a list of questions. Further please consider answering the following questions:

1) Can the authors include Top-5 accuracies for Tables 1, 2?


**Limitations:**

Briefly mentioned in Checklist 1(C).

**Strengths And Weaknesses:**

**Strengths:**
1) This paper is written well. It is easy to follow.
2) The improvements in terms of FLOPs and parameter reduction using VTC-LFC is impressive. VTC-LFC has potential to be useful in edge-intelligence applications (where ViTs are superior to CNN based architectures).

**Weaknesses:**
1) **The overall compression procedure is unclear**. I.e.: There is also a knowledge distillation component (Line 159). Can the authors show the overall algorithm (a pseudocode will be useful)?
2) **Only final top1 accuracy of VTC-LFC is insufficient to understand the significance of the proposed method / More analysis is needed**: This paper could significantly benefit from more analysis / statistics regarding channel pruning and token pruning. Given that LFS and LFE are scalar values, can the authors show the distribution of these values in Baseline / VTC-LFC ViTs? (A simple histogram could be useful).
3) **The cutoff point between low-frequency and high frequency for the experiments is not clear**: How is Figure 4 ( c ) obtained? Is it obtained by radial averaging of 2D Fourier spectrum similar to these deepfake works ( [1, 2, 3, 4] )?
4) The caption of Figure 3 is too plain. Please consider explaining the details as Figure 3 is very important in this paper.

Overall I enjoyed reading this paper. In my opinion, the strengths of this paper outweigh the weaknesses by a small margin. I’m happy to change my opinion based on the rebuttal.

=================

[1] Dzanic, T., Shah, K., & Witherden, F. (2020). Fourier spectrum discrepancies in deep network generated images. Advances in neural information processing systems, 33, 3022-3032.

[2] Durall, R., Keuper, M., & Keuper, J. (2020). Watch your up-convolution: Cnn based generative deep neural networks are failing to reproduce spectral distributions. In Proceedings of the IEEE/CVF conference on computer vision and pattern recognition (pp. 7890-7899).

[3] Chandrasegaran et al., 2021: "A closer look at fourier spectrum discrepancies for cnn-generated images detection." Proceedings of the IEEE/CVF Conference on Computer Vision and Pattern Recognition. 2021.

[4] Schwarz et al., 2021: "On the Frequency Bias of Generative Models." Advances in Neural Information Processing Systems 34 (2021).

---

> ### Author Response · Authors · 2022-08-02
> **Responses to reviewer 2SYW**
>
> **R3W1**: The pseudocode of the overall compression procedure is shown in Appendix A.1, where we illustrate the detailed procedures of pruning tokens and channels from the first block to the last block. The distillation component in line 159 is the KL divergence (in Eq.(3)) between two kinds of CLS tokens, one is the output CLS token when the input is the natural image and the other one is the CLS token corresponding to the filtered image.
>
> **R3W2**: Thanks for your suggestions. The distribution of LFS and LFE in baseline (original DeiT-Tiny) and VTC-LFC (our pruned DeiT-Tiny) ViTs are shown as follows. The distribution statistics is presented in the following tables (as figures are not supported here).
> | LFS |        | 0~5e-5   | 5e-5~1e-4 | 1e-4~1.5e-4 | 1.5e-4~2e-4 | 2e-4~2.5e-4 | 2.5e-4~3e-4 |
> |------|-------|:-:|:-:|:-:|:-:|:-:|:-:|
> | Density (%) | Baseline (original DeiT-Tiny) | 87.57     | 4.65        | 2.30        | 1.10        | 0.82        | 0.55 |
> | Density (%) | VTC-LFC (pruned DeiT-Tiny)  | 85.06     | 5.64        | 2.58        | 1.52        | 0.87        | 0.76 |
>
> | LFE |       | 0.28~0.29 | 0.29~0.3 | 0.3~0.31 | 0.31~0.34 | 0.34~0.35 |
> |------|-------|:-:|:-:|:-:|:-:|:-:|
> | Density (%) | Baseline (original DeiT-Tiny) | 9.69     | 10.86    | 10.92     | 9.94      | 8.35 |
> | Density (%) | VTC-LFC (pruned DeiT-Tiny)  | 9.26     | 10.59    | 10.89     | 9.93      | 8.43 |
>
> **R3W3**: The cutoff factor in our experiments is similar to the radial averaging of the 2D Fourier spectrum as in [1,2,3,4]. We will add the illustration in the revised version. Figure 4 (c) shows the performances of pruned models with different cutoff factors, and the model achieves the best performance with the value of 0.1. We have also visualized the filtered images and their corresponding Fourier spectrum in Appendix A.8.
>
> **R3W4**: Thank you for the suggestion. We have rewritten the caption of Figure 3 as below for a better illustration. The next version of our paper will be updated accordingly.
>
> Figure 3: Pipeline of Bottom-up Cascade Pruning. The compression is executed from the first block to the last one. In each block, the number of tokens is firstly reduced (red boxes) by selecting tokens based on the LFE-based criterion. When the performance drop induced by token pruning reaches a pre-set threshold, the number of tokens is fixed and the channel pruning (yellow boxes) is started. The channels are gradually removed according to their LFS until the accuracy drop reaches a pre-set threshold.
>
> **R3Q1**: Thank you for the suggestion. We have included the top-5 accuracies in Tables 1, and 2. The revised tables are as follows:
>
> Table 1: Comparison with state-of-the-art methods on ImageNet-1k. 'FLOPs $\downarrow$' denotes the reduction ratio of FLOPs. We report two versions with different parameter sizes for our method.
> |Method|&#124; |DeiT-Tiny| |&#124; |DeiT-Small| |&#124; |DeiT-Base| |
> |:-:|:-|:-|:-|:-|:-|:-|:-|:-|:-|
> | |&#124;Top1/Top5(%)|FLOPs $\downarrow$|Params|&#124;Top1/Top5(%)|FLOPs $\downarrow$|Params|&#124;Top1/Top5(%)|FLOPs $\downarrow$|Params|
> |Baseline |&#124; 72.2$/$91.1  | $-$ | 5.7M |&#124; 79.8$/$95.0 | $-$ | 22.1M |&#124; 81.8$/$95.6 | $-$ | 86.4M |
> |SCOP |&#124; 68.9$/-$ | 38.4\% | 5.7M |&#124; 77.5$/-$ | 43.6\% | 22.1M |&#124; 79.7$/-$ | 42.0\% | 86.4M |
> |PoWER |&#124; 69.4$/-$ | 38.4\% | 5.7M |&#124; 78.3$/-$ | 41.3\% | 22.1M |&#124; 80.1$/-$ | 39.2\% | 86.4M |
> |CP-ViT |&#124; 71.2$/-$ | 43.3\% | 5.7M |&#124; 79.1$/-$ | 42.2\% | 22.1M |&#124; 81.1$/-$ | 41.6\% | 86.4M |
> |EViT |&#124; $-/-$ | $-$ | $-$ |&#124; 78.5$/$94.2 | 50.0\% | 22.1M |&#124; 81.3$/$95.3 | 34.1\% | 86.4M |
> |HVT |&#124; 69.7$/$89.4 | 46.2\% | 5.7M |&#124; 78.0$/$93.8 | 47.8\% | 22.1M |&#124; $-/-$ | $-$ | $-$ |
> |IA-RED$^2$ |&#124; $-/-$ | $-$ | $-$ |&#124; 79.1$/$94.5 | 31.5\% | 22.1M |&#124; 80.3$/$95.0 | 33.0\% | 86.4M |
> |S$^2$ViTE |&#124; 70.1$/-$ | 23.7\% | 4.2M |&#124; 79.2$/-$ | 31.6\% | 14.6M |&#124; 82.2$/-$ | 33.1\% | 56.8M |
> |SPViT |&#124; 70.7$/$90.3 | 23.1\% | 4.9M |&#124; 78.3$/$94.3 | 28.3\% | 16.4M |&#124; 81.6$/$95.5 | 33.1\% | 62.3M |
> |VTC-LFC |&#124; 71.6$/$90.7 | 46.7\% | 5.1M |&#124; 79.4$/$94.6 | 54.4\% | 17.7M |&#124; 81.2$/$95.2 | 57.6\% | 63.5M |
> |VTC-LFC |&#124; 71.0$/$90.4 | 41.7\% | 4.2M |&#124; 79.6$/$94.8 | 47.1\% | 15.3M |&#124; 81.6$/$95.6 | 54.4\% | 56.8M |
>
> Table 2: Throughput of baselines and compressed models. 'Speed up' means the improvement in throughput. 'base' denotes the baseline model, and 'pruned' is the compressed model.
> |Model|&#124;Top1|&#124;Top5|&#124;Throughput|&#124;Speed up|
> |:-:|:-|:-|:-|:-|
> ||&#124; (base/pruned)|&#124; (base/pruned)|&#124; (base/pruned)|&#124; |
> |DeiT-Tiny |&#124; 72.2\%$/$71.6\% |&#124; 91.1\%$/$90.7\% |&#124; 2648.7$/$4496.2 |&#124; 69.7\% |
> |DeiT-Small |&#124; 79.8\%$/$79.4\% |&#124; 95.0\%$/$94.6\% |&#124; 987.9$/$1946.3 |&#124; 97.0\% |
> |DeiT-Base |&#124; 81.8\%$/$81.2\% |&#124; 95.6\%$/$95.2\% |&#124; 314.7$/$651.9 |&#124; 107.1\% |

---

> > ### Comment · Reviewer_2SYW · 2022-08-09
> > **Reply**
> >
> > Thank you authors for the great effort on the rebuttal. Authors have addressed all my concerns.
> >
> > In the revised version, please include the **distributions of LFS and LFE (results shown in the rebuttal) and consider explaining the cutoff-point between low-frequency and high-frequency.**
> >
> > Overall, this paper is interesting, well executed, and constitutes a good scientific contribution in my opinion. **I will increase my recommendation to Accept**. Good job on the rebuttal!

---

> > > ### Author Response · Authors · 2022-08-09
> > > **Reply**
> > >
> > > Thank you for the comments, we will include the distribution of LFS and LFE in the appendix and the explanation of the cutoff-point in the revised version.

---

### Official Review · Reviewer_KA3x · 2022-07-11

**Rating:** 7
**Confidence:** 5
**Soundness:** 3 good
**Presentation:** 4 excellent
**Contribution:** 4 excellent

**Summary:**

This paper introduces a model compression approach based on low-frequency components for vision transformers. Channel pruning based on low-frequency sensitivity, token pruning based on low-frequency energy, along with bottom-up cascade pruning scheme effectively reduce the model complexity while maintaining high model accuracy.

**Questions:**

1. How effective would this approach be when applied to other ViT models such as LV-ViT?
2. Would it be useful for window-based attention and hierarchical model structure such as Swin?

**Limitations:**

adequate

**Strengths And Weaknesses:**

strengths:
1. it proposes a novel approach for model compression with low-frequency component, by leveraging the intrinsic property of self-attention modules as low-pass filters.
2. it introduces the bottom-up cascade pruning framework for model compression.
3. it outperforms many SOTA vision transformer compression methods with higher accuracy at reduced model complexities

weaknesses:
it's unclear how effective this approach would be when applied to other ViT models such as LV-ViT.

---

> ### Author Response · Authors · 2022-08-02
> **Responses to reviewer KA3x**
>
> **R2W1&R2Q1&R2Q2**:We have evaluated the proposed methods on LV-ViT-S and Swin-Tiny. To simplify, we do not use the token labels in the original LV-ViT. Due to the token downsampling and the window shifting, we cannot apply token pruning to Swin models yet. Thus, only the channel pruning is adopted on Swin. We will study the token pruning for such window-based attention models in further works. Due to the time limit, the experiment has only run once. The results are listed as below. Our method outperforms the baseline on both models, showing that the proposed method is effective for different transformer backbones. We will conduct more experiments and update the results in the next version.
>
> | Model                 |&#124; Top1(%) |&#124; Top5(%) |&#124; FLOPs(G) |&#124; Param(G) |
> |-----------------------|---------|---------|----------|----------|
> | LV-ViT-S              |&#124; 83.2    |&#124; 96.3    |&#124; 6.5      |&#124; 25.8     |
> | NViT*+EViT (baseline) |&#124; 81.5    |&#124; 95.3    |&#124; 3.3      |&#124; 20.2     |
> | VTC-LFC (ours)        |&#124; 81.8    |&#124; 95.6    |&#124; 3.2      |&#124; 20.2     |
>
> | Model            |&#124; Top1(%) |&#124; Top5(%) |&#124; FLOPs(G) |&#124; Param(G) |
> |------------------|---------|---------|----------|----------|
> | Swin-Tiny        |&#124; 81.1    |&#124; 95.5    |&#124; 4.5      |&#124; 28.3     |
> | NViT* (baseline) |&#124; 79.4    |&#124; 94.7    |&#124; 3.3      |&#124; 17.1     |
> | VTC-LFC (ours)   |&#124; 79.8    |&#124; 94.8    |&#124; 3.3      |&#124; 17.1     |

---

### Official Review · Reviewer_ZeeK · 2022-07-11

**Rating:** 7
**Confidence:** 5
**Soundness:** 3 good
**Presentation:** 4 excellent
**Contribution:** 3 good

**Summary:**

In this paper, the authors proposed the pruning method with low-frequency for ViTs. The metrics named low-frequency sensitivity for channel pruning, and low-frequency energy for token pruning are proposed, where the low-frequency image and feature are introduced. Besides, a bottom-up cascade pruning is proposed to prune the channels and tokens gradually. The experiments show a good reduction ratio of FLOPs and throughput with less performance drop, and the effectiveness of each proposed components.

**Questions:**

1) In the section 4.1 Line 228, the pruned models DeiT-Tiny/DeiT-Small/DeiT-Base are finetuned for 300/150/75 epochs, respectively.
     - Why the number of epochs for the three models are different?
     -  How many epochs to finetune the pruned models for other pruning methods, like SPViT, S2ViTE? Is it comparable to that in this paper?
     -  It usually takes 300 epochs to train DeiT-Tiny on ImageNet-1k. Why the pruned DeiT-Tiny needs finetuning for 300 epochs?

2) In the section 3.4 Bottom-up Cascade Pruning, the model is pruned from the first block to the last one. Is there any comparison to the method from the last block to the first one?
3) In line 164, how to set the hyper-parameter λ in the experiment? Is there any ablation study on it?

**Limitations:**

Yes

**Strengths And Weaknesses:**

Strengths:
1. The authors introduced low-frequeny image and feature to prune the ViT models. It is the first work to compress the model in the frequency domain.
2. The improvement on the throughput is significant with less performance drop.
3. The formulas of the proposed pruning method are clear and correct.
4. The paper is well written.

Weaknesses:
1. The performance metric is not sufficient. The pruned model can be evaluated on ImageNet-Real/V2 and downstream classification datasets, like CIFAR-10/100.
2. In Table 1, there is no comparison on the number of the finetuning epochs.
3. In Table 5, the accuracy of the model without LFS and BCP after finetuning is close by 0.2 to VTC-LFC. The difference is too small to show the effectiveness of the proposed method.

---

> ### Author Response · Authors · 2022-08-02
> **Responses to reviewer ZeeK**
>
> **R1W1**: Thank you for the suggestion. We have evaluated our pruned models on ImageNet-Real and CIFAR-10. Due to the time-limited, we will continue to update more results in the future. The results are shown as follow:
>
> |Dataset|Model|Top1 (%)|Top5 (%)|FLOPs (G)|Params (M)|
> |:-|:-|:-|:-|:-|:-|
> |||base/pruned|base/pruned|base/pruned|base/pruned|
> |ImageNet-Real|DeiT-Tiny|80.1/79.5|94.4/94.2|1.3/0.7|5.7/5.1|
> ||DeiT-Small|85.7/85.4|96.9/96.7|4.6/2.1|22.1/17.7|
> ||DeiT-Base|86.8/86.2|97.1/96.9|17.6/7.5|86.4/63.5|
> |CIFAR-10|DeiT-Tiny|98.1/97.9|99.9/99.9|1.3/0.7|5.5/4.9|
> ||DeiT-Small|98.6/98.6|99.9/99.9|4.6/2.1|21.7/17.3|
> ||DeiT-Base|98.9/98.8|99.9/99.9|17.6/7.5|85.8/62.7|
>
> **R1W1&R1Q1**:
>
> 1) Thank you for the comments. The finetuning epoch of DeiT-Base/Small/Tiny is set to 75/150/300 for the trade-off between finetuning time and performance due to the limited GPU resources. In fact, DeiT-Base and DeiT-Small will achieve the best performance with 150 and 300 epochs, respectively. In our experience, larger models need fewer finetuning epochs to recover the performance because of their stronger model capacity. This conclusion is similar to that in the self-supervised pre-training task, where ViT-B needs to be finetuned for 100 epochs, while ViT-L needs finetuning for only 50 epochs. Thus, the pruned DeiT-Small/Dei-Base is only finetuned for 150/75 epochs while the DeiT-Tiny needs 300 epochs to be on par with SOTA performance.
>
> 2) The SPViT conducts pruning from pretrained DeiT models (300 epochs) and finetunes the pruned models for 130 epochs. Because SPViT only reduces FLOPs by 20%$\sim$30% (vs 40%$\sim$60% FLOPS for our models), it needs fewer finetuning epochs to recover the performance. S2ViTE trains and prunes the models from scratch for 600 epochs. Compared to S2ViTE, we use less epochs in total. The performance and training time with different finetuning epochs for three models are listed as following:
>
> |Model|75 epochs|150 epochs|300 epochs|GPU Days (A10)|
> |:-|:-|:-|:-|:-|
> ||Top1/Top5(%)|Top1/Top5(%)|Top1/Top5(%)|75ep/150ep/300ep|
> |DeiT-Tiny|70.1/89.9|70.8/90.2|71.6/90.7|2.3/4.6/**9.2**|
> |DeiT-Small|78.6/94.4|79.4/94.6|79.8/94.9|3.3/**6.7**/13.3|
> |DeiT-Base|81.2/95.2|81.3/95.3|81.3/95.1|**7.5**/15.0/30.0|
>
> **R1W3**: Thank you for the comments. There are two main reasons. (1) The baseline is strong. The original DeiT-Small already achieves 79.8% (upper bound), so improving the performance from 79.2 to 79.4 is a significant improvement with such a strong baseline. We can observe LFS achieve a larger improvement by 0.5% for a weaker baseline in Table 3. (2) As lines 270$\sim$277 demonstrated, for a fair comparison, we keep the same pruning ratios (determined by our VTC-LFC using BCP) of each block in all experiments. Thus, only the pruning order (block by block) determined by BCP will influence the results in Table 5. If we do not control the pruning rate, w/o BCP (equals to "LFS + LFE") achieves 78.7% with 55.0% Flops reduction in Table 3, the improvement from the BCP reaches 0.7% (79.4% vs 78.7%). The improvements can verify the effectiveness of the proposed modules.
>
> As BCP is not a major contribution in this paper, we haven’t included a lot of ablation studies for BCP. We will update the results in the revised version.
>
> **R1Q2**: The main reason for pruning from the first block to the last one is that the pruning in previous blocks will change the inputs of their subsequent blocks, which may influence the value of our proposed LFS and LFE. If the last block is pruned first, the selected channels and tokens in this block will need to be re-adjusted when former blocks are compressed. Instead, if the former blocks are pruned firstly, the inputs for the subsequent blocks are fixed. We compare both pipelines on DeiT-Small, and the results are shown as follow:
>
> |Pipeline|&#124;Top1 w/o FT|&#124;Top5 w/o FT|&#124;Top1 w/ FT|&#124;Top5 w/ FT|&#124;FLOPs|&#124;Params|
> |:-|:-|:-|:-|:-|:-|:-|
> |Bottom-up (from first to last)|&#124;68.3|&#124;89.1|&#124;79.4|&#124;94.6|&#124;2.1G|&#124;17.7M|
> |Top-down (from last to first)|&#124;68.2|&#124;89.0|&#124;78.9|&#124;94.5|&#124;2.1G|&#124;17.6M|
>
> It can be found that the bottom-up pipeline achieves better performance than the other pipeline, which demonstrates the significance of pruning from the first block to the last one.
>
> **R1Q3**: The hyper-parameter λ is set to 0.1 in the experiment. Here, we show the results of pruning 20% channels from DeiT-Small with different λ. Due to the time limit, only the results without finetuning have been obtained. It can be found that the model achieves the best accuracy when λ is 0.1. In addition, the proposed method is not very sensitive to the value of λ (λ>0).
>
> |$\lambda$|&#124;0.0|&#124;0.1|&#124;0.2|&#124;0.3|&#124;0.4|&#124;0.5|&#124;0.6|&#124;0.7|
> |-|-|-|-|-|-|-|-|-|
> |Top1|&#124;65.90|&#124;**69.76**|&#124;69.70|&#124;69.73|&#124;69.73|&#124;69.70|&#124;69.67|&#124;69.67|

---

> > ### Comment · Reviewer_ZeeK · 2022-08-09
> > **Good response**
> >
> > I am glad to see the additional experiments and details, which addressed my concerns. Thanks to the authors for the efforts in this discussion. After reading the comments of other reviewers, I think this paper should be got in.
> >
> > Moreover, since the compression of vision transformers emerges several new techs, there are many recent good papers studying this topic, pls consider discussing and comparing with the following works and other related ones:
> > - MiniViT: Compressing Vision Transformers with Weight Multiplexing
> > - TinyViT: Fast Pretraining Distillation for Small Vision Transformers

---

> > > ### Author Response · Authors · 2022-08-10
> > > **Reply**
> > >
> > > Thank you for the comments and suggestions, we will discuss the MiniViT, TinyViT, and other related ones in the revised vision.

---

### Meta-Review · Area_Chair_3paR · 2022-08-20

**Recommendation:** Accept
**Confidence:** Certain

**Metareview:**

The authors present a method to improve ViT efficiency by pruning channels and tokens using a selection mechanism that emphasizes low spatial frequency information. In particular they propose two measures: Low Frequency Sensitivity (LFS) and Low Frequency Energy (LFE).

1) LFS comprises two parts (Eq 3), the contribution between is controlled by a weighting hyper-parameter (though ultimately a Taylor approximation is used):
	a. The difference in the loss function of the model on the low-pass image with and without a removed weight.
	b. The difference in KL divergence of the class token before and after low-pass filtering with and without the weight in question.
2) LFE measures the proportionate energy of a token among the energy of all tokens after low-pass filtering (Eq 6), combined with a measure of the attention weights (Eq 8).

Pruning is carried out via a bottom-up cascade mechanism (Sec 3.4).

Performance is evaluated on a variety of model sizes on ImageNet 1K.

Pros:
- [R] First work to compress a model emphasizing low-frequency information
- [AC/R] Throughput improvement is significant with minimal performance drop.
- [R] Clear and correct formulas.
- [AC/R] Well written.
- [AC/R] Outperforms other compression methods.

Cons:
- [R] More thorough evaluation needed (ImageNet Real/V2 and CIFAR 10/100). Authors followed up and provided this requested evaluation data.
- [R] No comparison on the impact of the number of epochs. Authors supplied information.
- [R] No ablation on hyperparameter for LFS. Authors supplied information.
- [R] Unclear how the method might work with window based ViT such as Swin. Authors provided these additional experiments.
- [R] Unclear what is the cutoff for low/high frequency components. Authors provided ablation of varying cutoffs.

Paper has unanimous accept ratings. Authors addressed reviewer concerns, and reviewers complimented authors for a good job. AC recommends accept.

AC Rating: Strong Accept

**Award:**

Yes

---

### Decision · Program_Chairs · 2022-09-14

Accept